# Statistical Quality Inspection Methodology in Production of Precast Concrete Elements

**DOI:** 10.3390/ma16010431

**Published:** 2023-01-02

**Authors:** Izabela Skrzypczak

**Affiliations:** Faculty of Civil and Environmental Engineering and Architecture, Rzeszow University of Technology, Powstancow Warszawy 12, 35-959 Rzeszow, Poland; izas@prz.edu.pl

**Keywords:** precast concrete products, quality control, OC curve, AOQ curve

## Abstract

Today, prefabricated concrete elements are used in many construction areas, including in industrial, public, and residential construction; this was confirmed via questionnaire research. In the article, the prospects for precast concrete development are presented, and the factors determining the use of this technology are defined. Based on a review of the literature, it was shown that currently, higher-quality prefabricated elements are primarily created through the implementation of innovative materials and production technologies. For this reason, the lack of research regarding quality control in prefabricated elements based on statistical quality control is particularly noticeable. The quality control process is one of the most important distinguishing features in prefabrication due to the increasingly stringent expectations of customers; it helps to ensure that the desired durability of implemented constructions is achieved. Issues related to assessing the effectiveness of standard procedures presented in this paper were analyzed using statistical methods in the form of OC (operating characteristic) and AOQ (average outgoing quality) curves. Thus, a new approach was proposed because these methods have not been previously used in precast concrete. The shape of the curves obtained confirmed the significant dependence of the value of the acceptance probability on the defectiveness of production. In AQL control systems based on OC and AOQ curves, it is necessary to calculate the current average defectiveness, which should be treated as a basis for the decision to switch from one type of control (normal, tightened, or reduced) to another. In this respect, the standard requirements of quality control have been simplified, and it has not been considered necessary to determine the average defectiveness value in production processes. The examples included in this study, including the analysis of curb production data, clearly show the harmful effects of ignoring the actual process defectiveness. As a result of the calculations, it was found that the average actual defectiveness of the curbs produced could not be equated with batch defectiveness. The analyses carried out in this study prove that equating batch defectiveness with process defectiveness is not an appropriate approach, which was confirmed through the producer’s/customer’s risk analysis. The approach proposed in this study, the analysis of OC and AOQ curves, is an innovative solution in prefabrication and can be an effective tool for managing the quality of prefabricated products, taking into account economic boundary conditions.

## 1. Introduction

The existence of prefabrication plants and numerous successful construction projects realized with precast concrete elements for all or part of a building structure demonstrate that this technology is effective and economical [1]. In [2], the author contended that by employing industrialized and automated building system methods, the amount of manual labor on-site could be reduced, and the construction speed could be increased along with higher construction quality being achieved. The advantages of prefabrication, i.e., ensuring repeatability, increasing efficiency, and achieving lower unit prices compared to traditional technology, mean that recently, prefabricated concrete elements have been increasingly used. These elements are manufactured under controlled conditions and strict supervision, and their standard—ensuring the requirements for mechanical properties, accuracy, and quality are met—is becoming increasingly higher. Quality control is a key element in management at all levels of production. Quality control makes it possible to verify the characteristics of prefabricated elements and ensures their ability to meet the needs and requirements of the customer [2]. Customer requirements are most often specified in the contract specification.

On the other hand, the level of quality in terms of efficiency, usability, aesthetics, strength characteristics, or geometric features are formulated in the relevant industry standards. The purpose of quality assurance and quality control systems is to maximize the repeatability of the manufactured prefabricated elements, and to ensure that the standard requirements are met [3,4]. In [4], a precast segment (rings for tunnel linings) manufacturing process’s quality control processes are presented and discussed. Despite the quality control system, some areas of weakness were identified. These arose primarily due to non-conformance in the casting process and a lack of maintenance. A substantial number of non-conformances were associated with surface defects in the segments. A review of the QA/QC control system showed there were no inadequacies in the system, but certain aspects could have been improved. These include the workflow of remedial works, equipment maintenance, and staff’s working attitude. Amongst the factors that affect the maintenance of good quality, workers’ cost and skill level were considered. It was shown that higher-quality precast segments are attainable through the careful choice of materials, the use of better production tools in batching and casting processes, and strict quality control. By considering time and cost factors, constraints may have been added to the selection of materials for quality testing. The frequency of selective testing was once every 12 months for raw materials and reinforcement and one monthly cycle for steel molds and reinforcement cages. The authors of [4] suggested that the variation in the test result should be scrutinized according to other standard practices. The variation might have resulted from non-random sampling, probability causes, or other factors that were not readily detectable. A predictable pattern, such as a normal distribution, should become a parameter of the chance of variation. Tightened quality control is necessary to minimize the impact of cost and time constraints in the production of precast products.

The standards provide recommendations, guidelines, and procedures for users, designers, and manufacturers to ensure the desired level of quality is achieved. These include production, aesthetic, technological, and economic considerations. Thus, quality control is an integral aspect of the production of precast concrete elements. Great importance has always been placed on assessing the building elements quality and structures and improving the safety and reliability of implemented facilities [5,6,7,8,9,10,11]. As noted in the research conducted to date, an efficient quality control system is the most critical and important process link in the mass production of prefabricated elements [3]. Currently, prefabrication plants evaluate prefabricated manufactured elements in several stages specified in industry standards [12,13] and codes, e.g., [14,15,16,17,18,19,20,21,22]. In addition to the quality control of each component, the mechanical properties of finished prefabricated elements are assessed.

On the other hand, the key features that ensure the high quality of the final product are, in addition to providing strength and deformation parameters: the geometric dimensions, the location of the reinforcement, and the surface finish. It is also challenging to measure and evaluate quality characteristics in a highly reproducible and efficient manner. Therefore, innovative measurement and quality control methods have been proposed [23,24,25,26,27,28,29,30]. Usually, the first step is to assess the quality indicators on an ongoing basis concerning the properties of both the input materials and the finished prefabricated elements, i.e., strength and deformation parameters, as well as data related to the geometric dimensions, reinforcement positions, and surface quality. The next evaluation stage is the acceptance of the finished products, which includes selective tests to assess the already-finished, full-size prefabricated elements using destructive methods [31,32,33,34]. The necessity of moving from selective inspection to full factory inspection has been demonstrated in previous research [30,35]. According to the research described in [25,30], the selective control systems for prefabricated elements in the standards do not always take into account the variability in the production process and do not always ensure the appropriate quality and reliability of all of the components of the manufactured elements from a given batch of tests. Therefore, in [30], the quality control of finished prefabricated elements based on reliability was proposed. Software systems for quality control based on probabilistic algorithms were also proposed, which allowed the influence of the variability in process factors and controlled parameters on the properties of the output products to be taken into account. Methodology for the automated quality control of reinforced concrete prefabricated elements was proposed based on ensuring the reliability of manufactured concrete elements.

Quality-related aspects are fundamentals in concrete precast plants, as prefabricated elements are the basis of modern construction [1,34]. The production of prefabricated concrete elements requires the essential requirement to be met—quality—while ensuring the statistically needed minimum number of elements are tested with the minimum control costs. The fulfillment of these aspects is possible through the continuous implementation of statistical quality control procedures with the production process itself, as well as the properly designed and implemented selective quality control of ready-made prefabricated elements. It allows achievement by applying standard quality control procedures formulated in codes and by automating production processes. Therefore, to ensure the quality of the manufactured elements, e.g., in [6,24,25,34], it was proposed to automate construction processes that covers the project’s entire life cycle. However, decisiveness is required in quality assurance to ensure the correct implementation of the appropriate technology [26]. Many companies and countries have recognized the great potential of automation, resulting in various pilot projects, patent applications, and many automation and robotic solutions being implemented in the construction sector. Qualitative and quantitative studies regarding the limitations and prospects for using robotics in construction are presented in [27]. According to [28], the degree of automation in producing precast concrete products is more advanced than in any other industry. As concrete is the most widely used building material worldwide [35,36], well-developed quality control procedures already exist for this material in terms of design and production. The methods are reflected in the compliance criteria recommended in concrete codes. Currently, in principle, the entire concrete industry has started to implement automated optimization processes for the product and production. In [37], the interdependencies between the material, structure, and production of precast concrete elements were described, and optimizations—including ecological-, economic-, and quality-related optimizations—of the manufactured precast concrete were indicated as overarching strategies for possible enhancement. In addition to optimizing the materials used, the composition of the concrete mix, reinforcement, production, and quality should be considered in relation to costs and the minimum number of elements being tested [38].

As was found, e.g., in [39,40], most published articles regarding concrete prefabrication focus on a specific method of producing prefabricated elements, as well as defining the categorization of significant interdependencies in the production of precast concrete elements [37]. Higher-quality prefabricated elements are primarily created through innovative materials [41,42,43,44,45,46,47,48,49,50,51,52] and production technologies [53,54,55,56,57,58,59,60,61,62,63,64,65,66,67,68,69]. Reducing carbon dioxide emissions is the future of the construction industry; hence, the work on the development of materials based on waste is an essential aspect in the development of prefabrication [41,42,43,44,45]. In [41], to increase concrete toughness, the crushed rubber with sizes from 1 to 3 mm and 3 to 6 mm was replaced by 5%, 10%, and 15% sand. To compensate the degradation of the strength and improve the workability of the concrete, the combination of two additives of nano silica and metakaolin additives with optimum values was used. Moreover, the compressive strength, tensile behavior, and modulus of elasticity were measured and compared. The results indicate that the optimum use of nano silica and metakaolin additives could compensate for the negative effects of the rubber material implementation in the concrete mixture while improving the overall workability and flowability of the concrete mixture. For this reason, the importance of research regarding the quality control of prefabricated elements based on statistical quality control is particularly noticeable. As seen from the literature review, the problem of the number of samples used in research in the production of prefabricated elements is also important to the effectiveness and efficiency of quality control and economic procedures, and very few articles have been published on this subject. The quality control process is one of the most important distinguishing features in prefabrication due to the increasingly stringent expectations of customers; its use can also ensure that the desired durability of implemented constructions is achieved. Issues related to assessing the effectiveness of standard procedures presented in the paper were analyzed using statistical methods in the form of OC (operating characteristic) and AOQ (average outgoing quality) curves.

### 1.1. Advantages of Prefabrication

The use of precast concrete elements has many advantages compared to the production of individual elements on a construction site. According to [1,46], the advantages of the prefabrication process and prefabricated elements which establish the competitiveness of this solution in the construction market include:-The actual high quality of the product, which is produced in a controlled environment using standardized methods.-Advanced quality control that goes far beyond fresh concrete control.-Dimensional accuracy, ensuring the properties of both hardened concrete and the position of the reinforcement can be checked before being embedded in the element structure.-Factory production is independent of weather conditions and can be carried out independently of on-site construction works.-Design flexibility: For a designer, a significant advantage of prefabricated elements is their great architectural value. The material offers unlimited possibilities in its formation, and the color, texture, and detail can be modified. The material can be used to implement design ideas in various architectural styles.-Economy: An important advantage of precast concrete elements is the production process itself; it ensures a low level of material consumption (concrete and steel). The prestressed process saves up to 50% of steel and higher. In addition, the factory production process is highly developed. The techniques used in the prefabrication industry make it possible to provide a high-quality finished product that meets the project requirements in terms of cost and schedule.-Low negative impact on the environment: Precast concrete is an environmentally friendly material. It is made of natural materials. No toxic substances are produced during its production and use.

Within the advantages of prefabrication, the key advantage is the high real quality of the product, which is guaranteed by standardized production methods and advanced quality control. The process of creating prefabricated elements with appropriate quality takes place through the implementation of basic management functions in production processes, i.e., planning, organizing, and controlling all of the activities in such a way that the result is a product that meets the assumed final needs. Quality is not only created during the final processing stage of the product on the production line but is also “created” in the preproduction, production, and post-production phases, determining the interdependence of a number of activities and leading to the creation of the final product. Repeatability throughout the cycle enables production development, constantly adapting it to customers’ needs, and emphasizing the constant interpenetration of production and consumption [47]. Quality planning sets goals and quality requirements as well as requirements for the application of the statistical quality control system. The quality of a final product depends primarily on the method of its design (including the quality of input materials), the quality of the workmanship, or the conditions connected with the production technology, such as temperature, humidity, pollution, or shocks caused by operating devices.

Companies producing prefabricated concrete products on the construction materials market are expected to continuously increase their production capacity, maintain the high-quality level of the offered products, and conduct a flexible pricing policy. Achieving these goals is possible through quality control at every stage of production, from the selection of input materials to ready-made prefabricated elements. To stay on the market, prefabrication plants must not only guarantee the high quality of the manufactured goods but also sell them at attractive prices for the customer. That dependence leads to the necessity of estimating the costs of controls related to the introduction of a given acceptance plan before its application. Therefore, manufacturing companies must carry out the acceptance control of a batch of products by a plan that guarantees the assumed level of quality with the lowest possible control costs. Then, it is essential to implement statistical control and an appropriate test plan for manufactured prefabricated elements in prefabrication plants. The goal is essential during the implementation of the quality control of prefabricated reinforced concrete elements, which, due to the complexity of climatic conditions, constitute the basis of modern construction in many European countries, including Poland. The production of prefabricated reinforced concrete structures requires the provision of the basic conditions which will ensure their reliability with minimal expenditure. This engineering and economic problem must be solved to improve the control system during the production process and the quality control of the output products to ensure that they meet the consumer requirements [30]. 

Currently, precast prefabrication plants assess the operational integrity of the output products in several stages specified in the applicable codes. First, the current control of individual quality indicators is carried out (for the properties of strength and deformation materials, data regarding geometric parameters, and reinforcement). The final product acceptance stage includes periodic tests to assess full-size elements. As described in [30], the selective control system is not always economically effective, does not consider the variability in the production process, and does not ensure the quality and reliability of all the elements in the tested batch. The need to move from selective to full factory control has been found in studies, e.g., [48].

The most rational solution is to evaluate the acceptance test plans used in ongoing inspections about specific parameters of the manufactured prefabricated elements. Consequently, the evaluation of the recommended test plans in the codes for precast elements of reinforced concrete structures based on statistical methods is gaining importance in research. The operational characteristic curves proposed in the article and the curves of average defectiveness after inspection, constructed for codes’ acceptance plans, allow one to account for the influence of variability in process factors and controlled parameters of prefabricated elements (the number of elements to be tested or the defectiveness of manufactured elements) on the consumer properties of output products. Therefore, this article aimed not only to determine the discriminant power of the recommended attribute test plans in the codes for prefabricated elements but also to determine the effectiveness of the quality control of precast elements at the acceptance testing stage.

### 1.2. Development Prospects for Prefabrication in Poland

The economic market crisis in 2020 led to implementing methods that will improve the construction process. Above all, this is the result of the work carried out in parallel in many segments of the construction industry, which generates many problems. The most significant factors for construction companies were the limited availability of employees, the duration of construction investments, and the assurance of the appropriate quality of the facilities. They, along with emerging difficulties, solutions, and technologies limiting negative premises, have gained importance. Modern prefabrication is the solution to these challenges. Therefore, it is not surprising that an increasing number of actors in the construction market are looking more and more favorably toward prefabricated elements than in previous years; their increased use not only allows the implementation of works to be significantly accelerated but also allows the appropriate levels of quality and durability of the facilities to be guaranteed. Therefore, in the first stage of the research, to assess the condition and prospects for the development of concrete prefabrication in Poland, a survey was carried out among investors, architects, and contractors. The research included 15 respondents from each target group. Based on the survey, it was found that prefabricated concrete elements represent a well-known and relatively commonly used solution in the implementation of most types of investments [49,50,51,52]. On average, it was shown that three out of four respondents implementing industrial and warehouse construction projects use prefabricated concrete elements. The responses obtained from the respondents also indicate the good popularity of this type of technological solution in particular segments of the construction industry (Figure 1). However, the widespread use of prefabricated concrete elements in a given construction segment does not necessarily translate into a large scale and a large share of this technology in the context of the entire investment (Figure 1).

The factors that determine the use of precast concrete elements in a construction site are shown in Figure 2. The distribution of responses differed depending on the group of respondents (Figure 2).

The factor that determines the use of precast concrete technology is the reduction in construction times. This aspect was shown to be crucial for almost half of the respondents in this group (42%) (Figure 3). It was found that quality is a decisive factor for architects in choosing prefabricated technology. For contractors, work cost is a crucial factor.

Another important factor when choosing a prefabricated technology for investors is quality and technical parameters. The ease of assembly is another factor that determines the use of this technology. However, for investors, factors such as material costs and durability are practically just as important.

From the perspective of contractors, concrete prefabrication technology was very positively evaluated. As a result of the research, the contractors positively (good and very good) assessed precast concrete elements (Figure 4).

When asked whether the share of contractors using precast concrete products will increase, up to 27% of contractors responded positively (Figure 5). This relatively high degree of popularization of this type of technology will reduce the potential for a further upward trend in the future.

It was shown that the surveyed contractors have no doubts about the time savings that result from using prefabricated elements (Figure 6). In the opinion of the contractors, none of the elements mentioned in the study (Figure 6) can be realized faster using traditional technology. Only in the case of walls did a noticeable percentage of respondents (31%) believe that both technologies are comparable in terms of time. On the other hand, it was shown that the surveyed contractors have no doubts about the speed with which lintels, balconies, staircases, and stairs can be created. Everyone indicated that the use of prefabricated technology translates into time savings.

The issue of the mentality and preferences of market participants remains a significant barrier to the dissemination of this technology in Poland, which results from bad experiences and associations with concrete prefabrication from the 1950s to the 1970s. However, as our research shows, favorable changes have occurred in this regard, largely due to the positive ratings from those who have already used this technology. The promotion of investments made of prefabricated elements is significant to the increase in awareness and the better perception of this technology, which we should expect an increasing level of in the future (especially in terms of housing construction). Research shows that the advantages of prefabrication, especially the shorter construction times and the stable quality of the manufactured prefabricated elements, are reflected in the use of this technology. Prefabricated product manufacturers have been modernizing their technological lines for years, increasing not only the number of products and solutions but mostly improving their quality parameters by implementing the statistical quality control procedures recommended in the codes. Therefore, the next research stage is the analyses of the procedures’ statistical quality control used in the quality assurance of the prefabricated elements produced. The results obtained from the questionnaire are similar to the research results in [52].

Quality assurance is based on the idea of preventing quality defects. It is the next step in developing quality concepts after quality control, which is based on statistical methods and developed research plans.

### 1.3. Statistical Quality Control

Statistical quality control (SQC) may include statistical acceptance control (SAC) and refer to statistical product acceptance (SPA) or statistical process control (SPC) focused on the production process [53,54,55,56,57], which may be combined with control charts [53], for example, Cusum charts (method C in the control of concrete compressive strength according to EN 206 + A2: 2021-08 [14]) (Figure 7).

The aim of SQC, in addition to ensuring the required level of the tested feature of products, is to reduce the cost of control and, in some cases, even to enable control.

SQC mainly deals with issues related to statistical methods of receipt of piece products and current control carried out during production on a random basis, i.e., a representative part of the examined whole.

These methods make it possible to reduce the number of erroneous judgments about the quality of tested batches of products, making the probabilities of rejecting a good batch (meeting the quality requirements) low enough (the first type of error) and accepting poor-quality products low enough (the second type error), as well as protecting against excessive shortages in production.

Acceptance control due to the measured and calculated parameters and the method of evaluation is divided into [48,57]:-Control based on attribute assessment: Batches of products subjected to checks should be considered compliant if the number of non-compliant items in the tested sample does not exceed the qualifying number.-Control based on variable inspection: A batch of products subject to control is considered compliant if the number of quality statistics do not exceed the qualifying number k.

The discriminatory power (effectiveness) of different sampling plans can be assessed by comparing how they perform their function at different possible levels of quality. The practical difficulty in finding the perfect random sampling plan is that you cannot change the laws governing random events. Therefore, when choosing collection plans based on batch testing with defective items, a decision should be made regarding what risks may be incurred in each case, and this is most often an economic decision. In the case of batch acceptance, according to an attribute assessment, the value of the risk of acceptance of batches containing a certain percentage of defective elements is given by the operating characteristic curve of a given acceptance plan.

### 1.4. Statistical Acceptance Control According to the Attribute Assessment

Statistical acceptance control methods define random sampling and provide the rules of a procedure to qualify the quality of finished product batches. On their basis, we can consider a batch good or defective. In the latter case, it is possible to lower the element’s class or sort and reject the defective precast elements and in justified cases, destroy and recycle them.

One of the methods used is so-called acceptance plans. The simplest is a single (one-step) attribute assessment plan classifying each item as good or defective. The quality of the batch is then understood as the quotient *j = (n − k)/n*, where n is the number of simples in the batch with *k* defective items, and the batch defect is called *w = k/n*. A single test plan is a formal record specifying the size of a random sample taken at one time from a lot and the allowable k number of defective items in the sample; exceeding the allowable number leads to the lot being classified as bad. The number *k* is called the qualifying number, and the plan is denoted by the symbol *k‖n*, where *n* is the number of test results for precast elements [48,57].

To create such a plan, the acceptable quality level (AQL) that meets the inequality *w ≤ AQL* is determined in each batch. The probability that, in an *n*-element sample taken from a batch with a defect in w, there will be at most *k* defective pieces, can be calculated from the Bernoulli formula (RB—Bernoulli distribution) for independent samples (1) [57]:(1)$$P_{a}=\sum_{k=1}^{n}\left(\begin{array}{l} n\\ k \end{array}\right)w^{k}{\left(1-w\right)}^{n-k}$$

The acceptance probability ($P_{a}$) of a defective lot (*w*) in the *k*‖*n* plan is called the characteristic of a single plan, and the graph of dependence on (*w*) is called the operating characteristic curve (OC curve). An ideal would be a plan to ensure that all of the lots with defectiveness *w ≤ AQL* are accepted and rejected when *w ≥ AQL*. However, this is only possible with faultless 100% control (Figure 8).

The better—or more selective—the reception plan is, the closer the OC curve plot is to that shown in Figure 8 (steeper).

When constructing the OC curves, it is possible to assume a different distribution of the analyzed features/parameters of the precast elements; therefore, Equation (1) can be approximated by the formulas [57]:-Poisson distribution—RP (2):(2)$$P(w)=\sum_{k=0}^{n}\frac{\lambda \exp(-\lambda )}{k!}$$
where λ=nw.-Gaussian distribution—RG (3):(3)$$P(w)\approx \varphi \left(\frac{k-nw+0.5}{\sqrt{nw(1-w)}}\right)$$
where $\varphi (x)=\frac{1}{\sqrt{2\pi }}\int_{-\infty }^{x}\exp\left(-\frac{t^{2}}{2}\right)dt$ is a function of the normal distribution N(0,1).-Normal distribution—RN for a random variable *k* with *n* and *w* parameters (4):(4)$$N\left(nw,\sqrt{nw(1-nw)}\right)$$

While for a random variable $\frac{k}{n}$ → the distribution of RN has the form (5):(5)$$N\left(w,\sqrt{\frac{w(1-w)}{n}}\right)$$

Formula (5) can be used when the condition is met *nw*(1 − *w*) > 4.

-Pascal distribution (RPa) for a random variable *n* with parameters *k* i *w* (6):
(6)$$P(w)=\left(\begin{array}{l} n-1\\ k-1 \end{array}\right)w^{k}{\left(1-w\right)}^{n-k}$$-Gamma distribution—RGa with the scale parameter *b* = 1/*w* and the random variable *n* − 1 (7):
(7)$$P(w)=\frac{n-1^{\left(\frac{1}{w}-1\right)}e^{-\frac{\left(n-1\right)}{2}}}{2^{\frac{1}{w}}\Gamma \left(\frac{1}{w}\right)}$$-The χ^2^ distribution—RChi^2^ with 2*k* degrees of freedom of a random variable 2*w*(*n* − 1) (8):
(8)$$P(w)=\frac{2w{\left(n-1\right)}^{\left(\frac{n-2}{2}\right)}e^{-\frac{2w(n-1)}{2}}}{2^{\frac{n}{2}}\Gamma \left(2k\right)}$$

The main purpose of the random acceptance of products is to determine whether the batch from which the collected element comes meets the previously assumed quality requirements and, thus, whether it can be considered compliant with these requirements. If the tested element does not meet the requirements, the entire batch should be considered non-compliant, and the procedure of dealing with a non-compliant product should be implemented. Due to the evaluation of the examined features, the acceptance inspection can be performed by an attribute or by a variables inspection. Only the measurable features are assessed during the acceptance test, e.g., size, density, and strength, using the variables inspection. A batch of products is considered compliant if the so-called quality statistic does not exceed the qualifying constant *k*, depending on the specified value of the acceptable quality level (*AQL*) and the number of samples—elements taken for testing [58,59].

According to the recommendations of the code ISO 2859-0: 2002 [18], non-conformities (defectiveness) should be classified in terms of their validity. Typically, a division of non-conformities into more significant—class A—and less significant—class B—is used. More important non-conformities will be controlled more strictly. By definition, the *AQL* is “a quality level that corresponds to the worst tolerated average process level” [18]. The *AQL* is a parameter of the process scheme because the *AQL* value, together with the letter sign of the sample size, allows one to determine the test plan and the control scheme. It is assumed that the average process level should be less than or equal to the *AQL* value in order to not reject too many batches produced.

During production control, a prefabricated element for testing is taken from each manufactured batch. The batch is considered compliant if the number of non-compliant units does not exceed the qualifying number *A_c_* (9) [18,19,57]:*k* ≤ *A_c_*
(9)
where *k*—number of non-compliant items per 10 items taken for testing;

*A_c_*—constant qualifying or disqualifying number of non-conforming items.

The control plan and control procedures using variables inspection define, e.g., code ISO 3951:1997 [20,21], complementary to ISO 2859-1:2003 [19]. In order to use variables inspection in the sampling procedure, the following conditions must be met:-A series of product batches supplied by a specific manufacturer using the same production process repeatedly is systematically checked;-The analyzed feature of the product is measurable on a continuous scale;-The production is statistically regulated, and the tested feature of the product has a normal or close to normal distribution.

All of the above-mentioned requirements are met for the production of precast concrete products (e.g., concrete paving stones, concrete paving slabs, and concrete curbs).

To determine the acceptance test plan, the values of the acceptable defectiveness are used, i.e., the AQL limit of acceptable quality, amounting to 0.1 (analogous to ISO 2859-1:2003 [19]). Depending on the severity of the non-conformities, different levels of AQL are used, e.g., significant non-conformities (e.g., the strength of paving slabs and the strength of curbs) are classified as class 1 and assigned higher values than less significant non-conformities (e.g., the dimensions of slabs and the dimensions of curbs).

Acceptance tests are carried out assuming a certain level of control, strictly related to the size of the batch and the number of samples taken. The code ISO 3951:1997 [21] provides three general levels of control: I, II, and III; and four special levels: S-1, S-2, S-3, and S-4. If there are no specific provisions in the relevant standards, level II data are usually used. Where stricter control is required, level III should be used, and when less severe control is needed, level I should be used. Special levels of control are used when a small number of test specimens and a high risk of random control (e.g., in a destructive strength test) are required simultaneously. For example, such situations occur when inspecting precast concrete products. The number of test samples can also be reduced by applying transition conditions between the different control levels.

Concerning the adopted test plans, it is possible to determine the dependence of the average defectiveness after the inspection of the defect and before the inspection, i.e., an actual defect in manufactured prefabricated elements (with a decreasing quality level). This relationship is called the average defectiveness curve after inspection (AOQ curve—average outgoing quality curves [56,58,59]. These curves can present the risk associated with acceptance plans for the quality control of the concrete used for production and the ready-made prefabricated elements.

Average outgoing quality limit (AOQL) can be treated as a criterion for the selection of a research plan, thus verifying the correctness of the adopted acceptance plans. Furthermore, the AOQL contains very important information for the recipient/consumer regarding the maximum defectiveness that can be expected with successive receipts of a number of batches over a long period of time based on the adopted acceptance inspection criteria. The practical significance of AOQL is that its value can be equated with the quantile level for the verified property (feature), and thus reflect the quality of the manufactured precast elements.

In AOQ charts, an extreme expresses the AOQL value. The coordinates of the AOQ plot can be determined based on the operational characteristics curve (OC) according to the formula [56,57]:AOQ = *w*·*P_a_*
(10)
where AOQ—the average defectiveness after inspection;

*w*—the defectiveness before inspection;

*P_a_*—the probability of acceptance of the batch of precast elements with defective *w.*

When verifying the correctness of the compliance criteria and the quality of the batch of precast elements, one can refer to two defectiveness values: 5% (AQL—acceptable quality level) and 10% (LQL—limiting quality level), for which L. Taewre [60] proposed formulas for boundary curves for three areas: controlled, unsafe, and uneconomical. In the case of concrete quality control, the determination of the impact of compliance control on the assumed concrete class requires the determination of the statistical quantile of the concrete compressive strength distribution after the compliance control. The quantile estimation for the compressive strength requires the determination of the concrete’s defectiveness after a compliance check. Such inference can be made using AOQ (average outgoing quality) curves (Figure 9). 

AOQ curves are also constructed for acceptance plans, i.e., for plans used in the acceptance of precast concrete elements. AOQ curves are curves obtained for the state after inspection; therefore, these curves make it possible to assess the discriminant power of acceptance criteria or compliance criteria in the case of concrete quality. Using AOQL for compliance control, the concrete compressive strength can be identified with the quantile defined for the concrete class, i.e., the quantile of the characteristic strength. According to EN 206 + A2: 2021-08 [14], if the industry standards do not define otherwise, the value of the permissible defectiveness after the inspection performed can be assumed to be at the level of 0.05 (11):(11)AOQL=0.05

## 2. Materials and Methods

### 2.1. Code Quality Control Procedures for Precast Elements

Contemporary codes for precast concrete products recommend acceptance control using statistical methods. In the case of prefabricated concrete products, an additional legal requirement [13] is the obligation to use CE marking for most of these products. This obligation is connected with the necessity of, among others, determining the type of product with the determination of its functional properties and conducting factory production control (FPC). Factory production control means “permanent internal control” carried out by the manufacturer to ensure the required product properties (both identification and use) [12]. FPC covers not only direct product control but also all aspects related to the production and supervision of this production [14,48,57], i.e.,:-The preparation stage of production, which includes, among others, the purchase and control of raw materials;-The production stage, including its supervision, control, and testing according to a predetermined plan;-Procedures for dealing with a non-conforming product;-The supervision of machinery and equipment, as well as equipment needed for inspection and testing;-The determination of the requirements for the competence of personnel;-Marking and securing products during storage and transport;-Corrective actions in the event of any non-conformities.

All of these activities should be properly documented. Therefore, the manufacturer should carry out controls and tests in all the stages of product manufacturing in accordance with the established frequency, which results from the technical specifications for a specific product and its production conditions. The frequency and number of samples taken for testing, and thus the control cost, are influenced by the level of control, among other things.

In addition, the subject standards for precast concrete products (e.g., PN-EN 13369:2018-05 [17]) emphasize that if the manufacturer has a quality management system in the company compliant with ISO 9001:2015-10 [22] and takes into account the requirements of the subject codes, it meets the requirements for implementing and conducting factory production control. On the other hand, the ISO 9001:2015-10 code [22], in chapter 9 on the evaluation of the effects of action, recommends the use of statistical methods for the analysis and evaluation of data, including for the conformity assessment of offered products (point 9.1.3 of code ISO 9001:2015-10 [22]). As can clearly be seen, statistical product quality control falls within the scope of obligatory factory production control, but it is only one of the activities leading to the obtainment of a product compliant with standard requirements and a good-quality product that will quickly attract customers.

### 2.2. Attribute Acceptance Plans for Selected Kinds of Precast Elements

Industry codes recommend using attribute acceptance plans for all precast concrete products [14,15,16,17,18,19,20,21,22]. In the case of the quality control of prefabricated concrete products, acceptance control always starts with normal control, and then, when the production level is good enough, reduced control can be applied, which involves fewer test samples and a lower qualifying constant k. Regarding prefabricated concrete products, the subject standards recommend that the qualifying constant k remain unchanged, but the number of samples can be reduced. Figure 10 shows the transition conditions between the levels of strength control levels for precast concrete elements according to the recommendations of EN 13369:2018-05 [17].

Each transition from a more severe to milder-level results in a reduction in the amount of control during the production of precast concrete elements.

Transitions between individual levels of control are formulated in the subject codes dedicated to specific precast products, e.g., in accordance with EN 1339:2005 [15], the control of paving slabs includes: the visual inspection of the product’s appearance, the measurement of the shape and dimensions of prefabricated slabs, the examination of wear layer thickness, strength determination to bending and breaking loads, and the determination of resistance to freezing/thawing with the use of de-icing salt (in resistance class 2). For each of these features, the EN 1339:2005 code [15] specifies the frequency and number of samples taken for the above-mentioned tests and the so-called conditions for the transition between the levels of control of the above-mentioned product characteristics. For example, when testing the flexural strength of concrete paving slabs with nominal lengths and widths of less than 300 mm, eight slabs of the strength family should be taken from the production machine on the day of production under normal inspection. During reduced control, this number is reduced to 4, and in additionally reduced control—to 2. In the event of disruption to the production process and the introduction of more restrictive control, the number of slabs is doubled with normal control and amounts to 16 units. Each transition between control levels is associated with a reduction or increase in the costs of conducting this control.

However, in the case of control and acceptance plans for the strength of concrete curbs, the EN 1340:2004 [16] code limits the testing plan of two levels of control (Figure 11).

For features other than strength, quality control covers three levels as standard (Figure 12).

### 2.3. OC and AOQ Curves Constructed for Acceptance Plans of Selected Precast Elements

In all the AQL-based test plans, normal inspection acceptance criteria are selected to protect the manufacturer from rejecting batches with acceptable quality for the inspected precast elements of the batch. However, in most test methods, the producer’s risk that such batches will be rejected varies depending on the plan adopted. Producer’s and customer’s risk analysis was performed regarding various permissible AQL defects, the sample number of n, the qualifying constant Ac, and various distributions of the analyzed feature. It was found that the customer’s risk that a batch with a quality worse than the AQL will be accepted is much greater with a small sample size, which was confirmed using the values of the dependence of the acceptance probability on the defectiveness shown in Figure 13 and the values in Table 1. The analyses were carried out by constructing OC curves for 5 types of distributions: RB—Bernoulli distribution; RP—Poisson distribution; RN—normal distribution; RCh^2^—χ^2^ distribution; and Gamma distribution—RG.

To assess the defectiveness after the quality control of the manufactured precast elements, AOQ curves were constructed for the acceptance plans 1‖5 and 1‖16. These curves were also constructed for 5 distribution types: RB—Bernoulli distribution; RP—Poisson distribution; RN—normal distribution; RG—Gamma distribution; RChi^2^—χ^2^ distribution (Figure 14).

### 2.4. Case Study of Concrete Curb Units

Styrobud of Podkarpacie Province in Poland provided the data for the analysis. Quality analysis was carried out regarding the continuous production of concrete curb units. In accordance with the recommendations of the EN 1340:2004 code [16], the test covered 8 curb units. The curb units had production dimensions of 998mm × 300mm × 198mm. The geometry and bending strength were evaluated. The description of the element’s geometry was adopted following the standard (Figure 15). 

The curbs belonged to quality class 2 (marking T), for which the characteristic bending strength is 5.0 MPa, and the minimum bending strength is 4.0 MPa. An alternative method was used to evaluate the geometric dimensions and the bending strength. Each of the requirements specified in the standard regarding geometry and bending strength were met by all the curbs, due to which, both the sample and the production lot from which they were taken were considered compliant with the requirements of the standard (Figure 16 and Figure 17). 

No curbs were shown to have a bending strength lower than the characteristic value of the declared class T (Figure 18).

Analyses of the geometry and bending strength were also carried out using 80 results (eight measurement cycles). The defectiveness was calculated in terms of the bending strength of the tested curbs—w = 0.04. Descriptive statistics for bending strength were determined (mean value—5.6 MPa and standard deviation—0.4 MPa), and the characteristic value of the bending strength was determined, which was 5.0 MPa. Based on the determined parameters, the characteristic value of the bending strength was calculated, which was slightly higher than the minimum characteristic value specified in the standard 5.3 MPa > 5.0 MPa. It was found that the requirements for the verified batch of curbs were met. However, the production defectiveness in terms of the height of the curbs, which was estimated based on 80 curb units, did not meet the requirements. The estimated defectiveness, w = 0.12, was greater than the permissible defectiveness of 0.10 (w = 0.12 > 0.10). The obtained values of defectiveness were lower than the limit values of the quantiles given in the EN 1340:2004 code [16] for bending—w = 0.03 < 0.05—and geometric dimensions other than height. The OC and AOQ curves were also constructed for the considered test plan 0‖8, and the acceptance probabilities of the tested curbs were determined for the geometry P_a_ (0.05) = 0.564 and the bending strength P_a_ (0.05) = 0.705 (Figure 19).

The evaluated batch of curbs, in terms of geometry and bending strength, could therefore be considered as products that meet the requirements of the EN 1340:2004 code [16]. However, doubts were raised in terms of the actual defectiveness of the curbs produced; therefore, the OC and AOQ curves were constructed, and the producer’s and the customer’s risks were estimated. In the analyzed case, the customer’s risk was 0.308, and the producer’s risk was 0.436. These values indicate how important the ongoing control of the actual defectiveness is. Thus, it was shown that using single simple plans according to the codes to evaluate small sets of test results (fewer than 15 elements) does not provide the trade-off between customer risk and manufacturer risk that is required in the development of reasonable quality plans. Based on the analyses carried out, it is recommended that, in addition to applying standard procedures to control individual batches, the actual defectiveness of the precast elements should be assessed. In the case of the inspection of a produced batch of precast elements, it is common practice that elements rejected during production are not included in the inspection, because the average process effectiveness is not calculated. In some cases, the number of defective elements identified prior to batch testing may justify the acceptance or rejection of the batch.

## 3. Results and Discussion

In engineering practice, the recognition of a material’s compliance with the specification is decided in the adopted plan for statistical quality control. It is a standard approach based on binary criteria (met/unmet). The action is of particular significance in the case of doubts concerning the quality of precast elements, in which the material quality is closely linked with the structure’s safety and reliability. It is essential to adopt an appropriate sample quantity when assessing quality.

The calculation of OC curves is based on the assumption of random sampling; however, obtaining a small number of non-biased samples from a very large batch is difficult. Larger numbers of batches with steeper OC curves allow for a more sensitive distinction between good and bad batches; the larger the batch, the more significant this distinction can be. The shapes of the OC curves in the acceptance plan mainly depend on the number of precast elements in the batch, the acceptance criterion, and the type of the adopted distribution (Figure 13).

The probability of accepting precast concrete products on the basis of an assessment of attributes with the assumed permissible defectiveness (AQL), the determined number of precast elements in the batch (n), and the qualifying number (A_c_) is variable and increases with the increasing number of samples (Table 1). As the number of samples increases, the OC curves become steeper.

It was found that the OC curves constructed for the Bernoulli, normal, and Poisson distribution and for the 1‖16 criterion equally protect the buyer against accepting batches with a defect of 0.05 (Figure 13, Table 1). It was shown that a test plan for normal distribution much more effectively protects the manufacturer against rejecting batches with a defect greater than 0.05. Several rejection values for batches with defect levels of 0.01, 0.05, and 0.10 in Table 2 confirm this fact.

When determining the acceptance probability, it is very important to assume the distribution of the measurable parameters/features and that the test result of the prefabricated element is random. It is, therefore, advisable not to stop at the one-time examination of the distribution form of the feature/parameter one is interested in but to examine as many independent elements as possible to be sure about the type of distribution form of a given feature/parameter so that it is possible to determine the effectiveness of the acceptance plan constructed for the assumed type of distribution.

However, when assessing defectiveness after the tests of the manufactured precast elements based on the constructed AOQ curves for the 1‖5 acceptance plan, it can be seen that only for the Chi^2^ distribution for the defect of the controlled area of 0.05–0.1 was the defectiveness of the assessed prefabricated elements lower than the value of 0 or 0.05. Additionally, in terms of the Gamma distribution, the pre-inspection defect at the level of 0.07 met the post-inspection defectiveness requirement of 0.05. In the case of the normal, Bernoulli, and Poisson distributions, the defect after the acceptance inspection of precast elements with defects up to 0.05 corresponded to the defect after the inspection, also with the value of 0.05 (Figure 14). However, in the case of the application of the 1‖16 acceptance plan and the defectiveness of the controlled area 0.05–0.1 and the adoption of the Chi^2^, Gamma, and normal distributions, the defectiveness obtained after the acceptance inspection of the assessed prefabricated elements was lower than the value of 0.05. In the case of the Bernoulli and Poisson distributions, the defectiveness after inspection was slightly higher and amounted to 0.051 and 0.052, respectively (Figure 14). Different values of the producer’s and consumer’s risks estimated based on OC curves are shown in Table 3.

Table 3 presents the OC curve results for producer’s risk α and consumer’s risk β with the desired values of AQL = 0.05 and LQL = 0.10. The effects of increasing the sample size on the OC curve while keeping acceptance number c constant are shown in Table 1 for different control plans: normal, tightened, and reduced. Increasing n while keeping c constant increased the producer’s risk α and reduced the consumer’s risk β. Raising the acceptance number for a given sample size increased the risk of accepting a bad lot β. An increase in the acceptance number from c = 1 to c = 2 increased the probability of obtaining a sample with two or less defects and, therefore, increased the consumer’s risk β. Thus, to improve single-sampling acceptance plans, management should increase the sample size n, which reduces the consumer’s risk β, and increase the acceptance number c, which reduces the producer’s risk α. The comparison of data in Table 1 shows the following principle: increasing the critical value for an acceptance number c while keeping the sample size n constant decreases the producer’s risk α and increases the consumer’s risk β.

The analysis of the OC and AOQ curves shows that it is possible not only to quantify the probability of acceptance of a batch of manufactured elements but also to quantify the manufacturer’s and recipient’s risks and defectiveness after the acceptance inspection of precast elements, and also to select acceptable risk levels and defectiveness values after the inspection by adjusting the sample size and/or by defining the specification limit, i.e., the permanent acceptance—Ac.

The value of the estimated acceptance probability related to the quality assessment of precast concrete products according to the attribute assessment formulated in the subject codes depends on the adopted distribution, assumed admissible defectiveness, assumed number of inspected elements, and the qualifying constant, and it is:Variable and increases with the increase in the number of samples;Depends on the level of control and the type of distribution adopted (be sure about the type of distribution of the feature under consideration).

The OC curves became steeper with the increase in the number of samples; therefore, adopting a larger number of elements for testing with tightened control is justified. The shapes of the constructed OC curves differed depending on the adopted type of distribution and the number of test elements. The differences in the values of the estimated acceptance probability were especially noticeable with the defectiveness up to 0.1.

The AOQ curves made it possible to determine the average value of the defectiveness after the performed acceptance inspection. Defectiveness after inspection depends on the number of samples, the type of distribution, and the defectiveness of the assessed prefabricated elements before the control.

The larger the number of samples, the more clearly there were differences between batches of different quality. It was shown that a larger number of samples more effectively protects the customer from accepting defective batches, and it also protects the manufacturer from rejecting batches of precast elements that meet the requirements.

The numbers of samples recommended in the codes are the minimum values; therefore, the proposed statistical methods enable the adoption of the number of samples that ensure the safety and reliability of the constructed structures. The number of samples can be assumed based on the shape of the OC or AOQ curves and the type and reliability class of the building structures according to EN 1990 [70], i.e., RC1, RC2, and RC3. According to the literature [54], the minimum statistically justified number of samples is six.

## 4. Conclusions

In this study, the literature regarding quality assurance and quality control in the production of precast elements was reviewed. The advantages of prefabrication were discussed in detail, and the state and prospects for the development of this technology in Poland were presented. The survey determined the factors that determine the development of prefabrication in Poland by investors, architects, and contractors. For investors, the main factor determining this technology’s choice was the time of investment implementation. In contrast, the main factor for architects was the quality and technical parameters.

By verifying the advantages of prefabrication, it can be stated that the key advantage of prefabrication is the high real quality of the product, which is guaranteed by the use of standardized methods to ensure and control both production and ready-made prefabricated elements.

European codes recommend the use of quality control procedures that ensure the implementation of a quality system in precast concrete plants. In addition, they recommend various measures to ensure quality consistency throughout the production process. These include the non-destructive and destructive testing of both materials and components, geometric dimensional checks, visual inspections, etc.

The subject standards allow tests with attribute methods (which determine whether a certain product feature is compliant or not) and with variables inspections (in which product features are strictly supervised and the mean value and standard deviation are measured). In some cases, the measurement of a given feature can be carried out with both methods. After meeting strictly defined requirements, this decision rests with the manufacturer. Acceptance based on variable inspection is more accurate and requires more research, and the calculations are more complicated. For this reason, attribute methods are often used to assess less important features, while inspection via variables is used to qualify more important features, e.g., compressive strength. The acceptance inspection according to the attribute evaluation, as according to the quantitative evaluation, enables the risk of the producer and the recipient to be balanced by estimating the probability of acceptance and adopting an appropriate control plan.

Acceptance sampling involves accepting or rejecting a unit (or batch) of goods. The design of the acceptance sampling process includes decisions about sampling versus normal inspection, attribute versus variable measures, AQL, α, LQL, β, and sample size. In precast production, management selects the plan with the code requirements (choosing sample size n and acceptance number c), and using an OC curve or AOQ curve, the effectiveness of acceptance plans formulated in the code can be assessed. If the sample size n is increased, with c, AQL, and LQL fixed, the OC curve would change so that the producer’s risk α increases while the consumer’s risk β decreases. Furthermore, with an increase in the critical value c, and with n, AQL, and LQL fixed, the probability of the producer’s risk α would decrease, but the probability of the consumer’s risk β would increase.

The analysis of the OC and AOQ curves carried out in the research based on alternative assessment is an innovative solution in prefabrication and can be an important tool for managing the quality of prefabricated products. The acceptance of a produced batch of prefabricated elements is based on a sample taken, and the defectiveness of a sample is identified via the defectiveness in the production process. However, as we know, this is not accurate. The actual defectiveness in the manufactured elements differs from the defectiveness in the sample taken, hence the different values of the acceptance probability depend on the defectiveness in the production.

In the AQL control systems proposed in the standard, based on the OC and AOQ curves, it is necessary to calculate the current average defectiveness in the process, which should be estimated based on samples from at least the last 10 batches. The value of the average defectiveness should be treated as a premise for the decision to switch from one type of control—normal, tightened, or reduced—to another. The standard simplifies the requirements governing such control changes and does not consider it necessary to calculate the average defectiveness value in the process, which does not seem to be justified. Passage for each quality level was limited to the inspection of individual batches, disregarding actual process defectiveness. The given example concerning the analysis of the defectiveness of produced curbs shows the estimation of the average defectiveness in the process. Based on the data, it was found that the average actual defectiveness of curbs in production may be greater than the estimated batch defectiveness. Values of average defectiveness higher than the admissible values were obtained for the height of the curbs. The verified batches met the standard requirements for the assessed batches but did not meet the requirement regarding production defectiveness. Equating batch defectiveness with process defectiveness is, therefore, not the correct approach, which was confirmed in the example provided. The analysis of the OC curves for tightened control showed that batches with clearly worse qualities than the AQL still had a fairly good chance of being accepted. It may therefore be desirable to supplement the stringent inspection criteria with some less formal requirements based on current estimations of the average defectiveness in the process. Based on the analyses carried out, it can be concluded that the requirement to calculate the average defectiveness in the process at regular intervals may have advantages. It is advisable for both the producer and the customer to know whether the quality is worse or better than the AQL value and whether the quality tends to improve or deteriorate. Note that the average defectiveness in the process calculated over a series of samples was simply the total number of defective items observed divided by the total number of items in the samples tested.

A review of the standard quality control procedures recommended in the codes and the evaluation of their performance using OC and AOQ curves showed no imperfections in the recommended quality control procedures. Based on the conducted analyses, it was found that the effectiveness of the quality control system concerning the selective quality assessment of precast elements depends on the adopted type of distribution of the tested feature and the adopted number of elements for testing. Quality control becomes less effective when a small number of samples is used, and acceptance plan control becomes very critical when many samples are used. Quality control becomes less effective when small samples are used for the acceptance plan.

The traditional approach to assessing the quality of precast products is through experiments and destructive testing, which are both time- and resource-consuming. The proposed statistical-approach-based method for quality assessment can be used to overcome these limitations. The suggested method may be employed in precast production and applied to predict the quality of precast elements by reducing, i.a., the numbers and scope of testing. The application of the proposed procedure combined with the use the statistical, fuzzy, or artificial neural network methods and modern measurement technology can ensure the reliable assessment of the quality of precast products.

## Figures and Tables

**Figure 1 materials-16-00431-f001:**
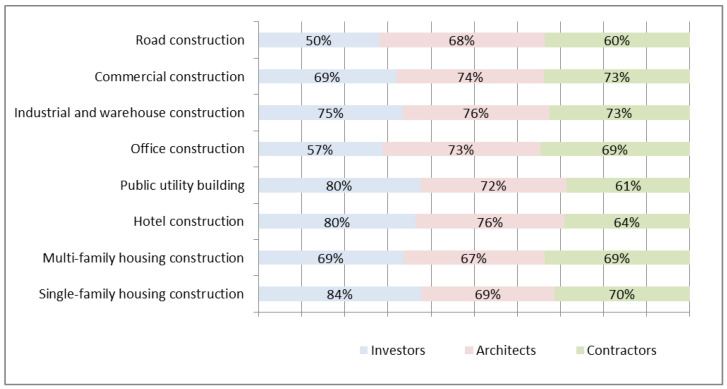
The use of prefabricated concrete elements during the implementation of the investment.

**Figure 2 materials-16-00431-f002:**
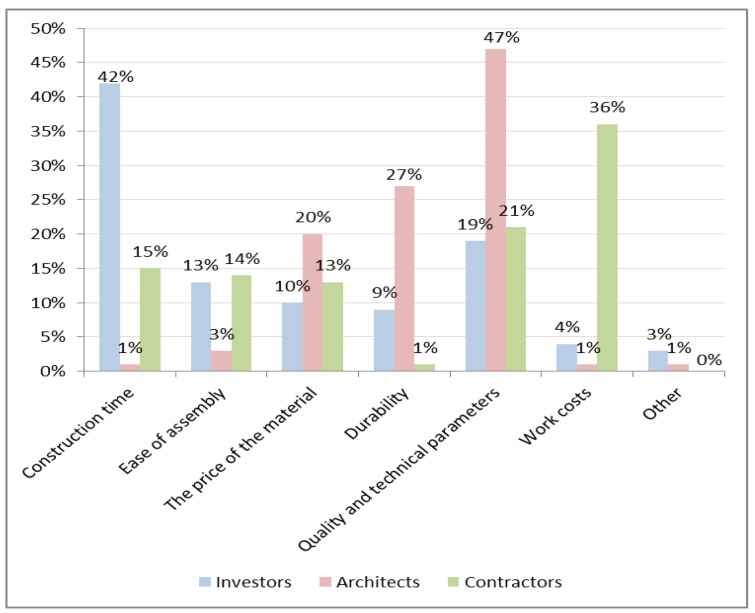
Factors that determine the choice of precast concrete technology.

**Figure 3 materials-16-00431-f003:**
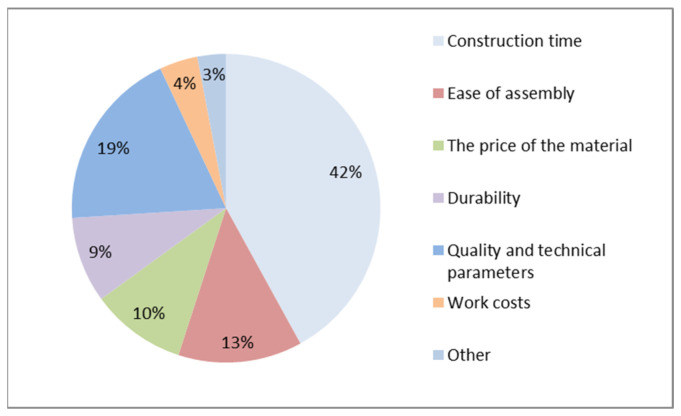
The most important factor when choosing the technology of precast concrete elements in the opinion of investors.

**Figure 4 materials-16-00431-f004:**
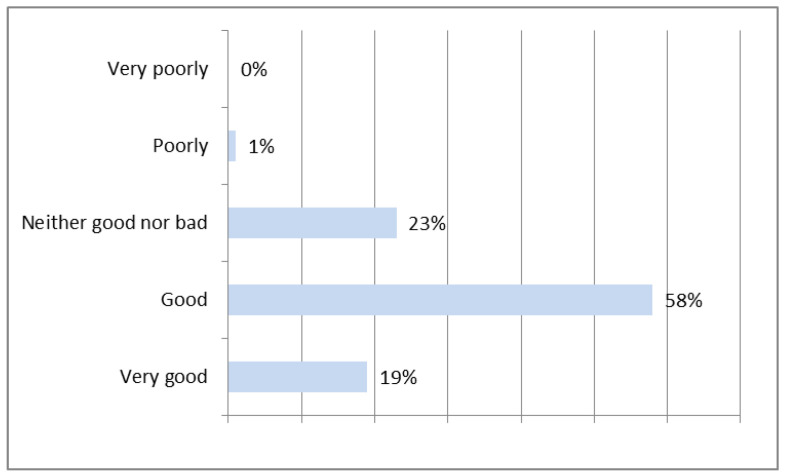
Evaluation of precast concrete elements—contractors.

**Figure 5 materials-16-00431-f005:**
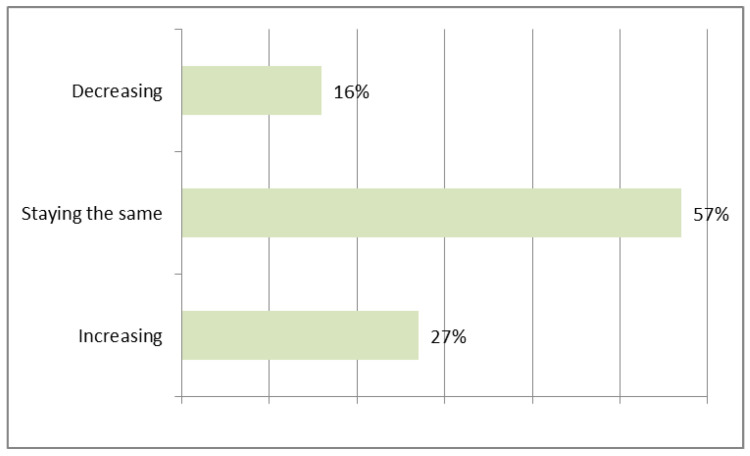
Expected change in the share of contractors using precast concrete.

**Figure 6 materials-16-00431-f006:**
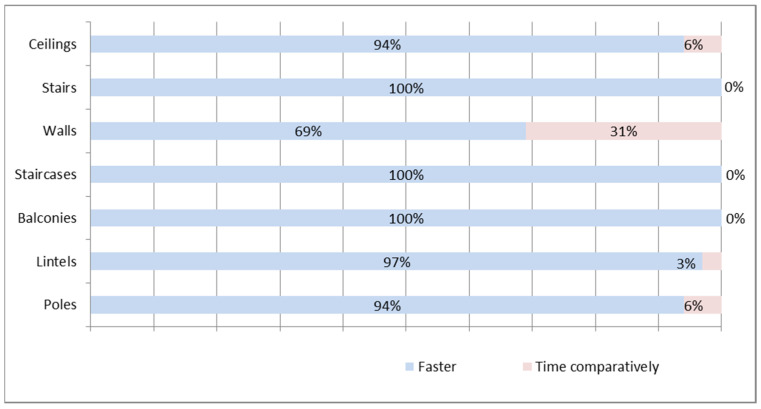
Implementation time—prefabricated technology and traditional technology—contractors.

**Figure 7 materials-16-00431-f007:**
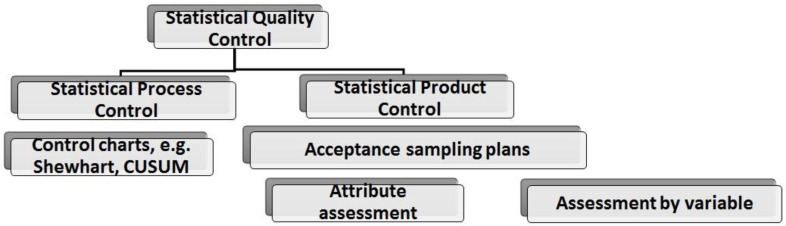
Scheme of quality control.

**Figure 8 materials-16-00431-f008:**
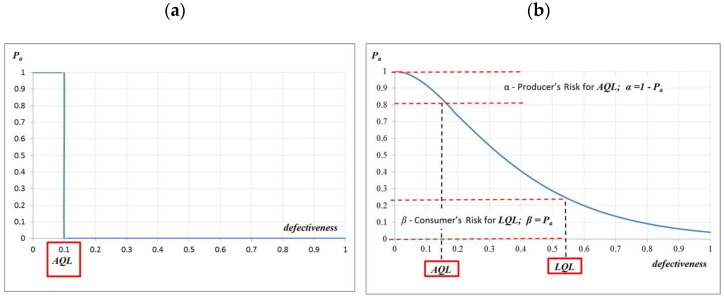
OC curve: (**a**) at 100% control and (**b**) with producer’s and customer’s risks.

**Figure 9 materials-16-00431-f009:**
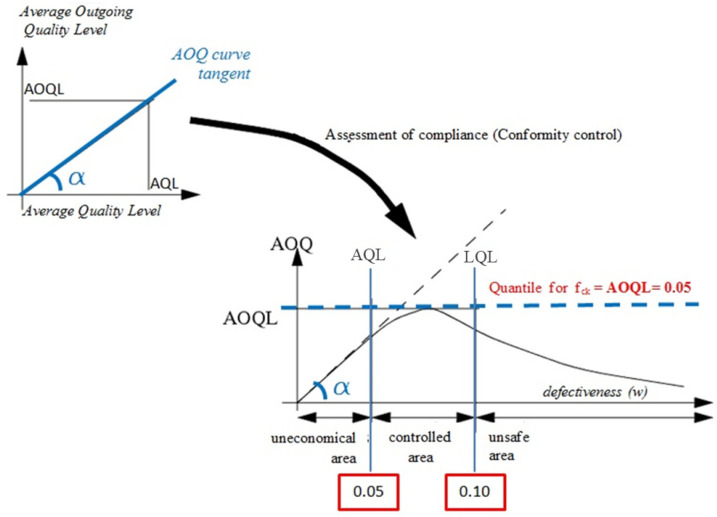
Diagram of the relationship between the defectiveness before and after quality control for compressive strength of concrete.

**Figure 10 materials-16-00431-f010:**
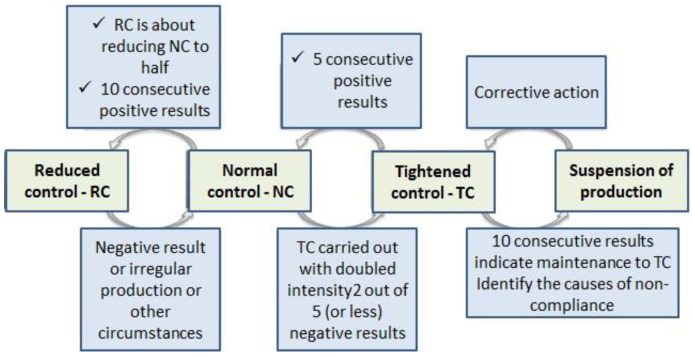
Conditions for the transition between the levels of strength control for precast concrete elements according to EN 13369:2018-05 [17].

**Figure 11 materials-16-00431-f011:**
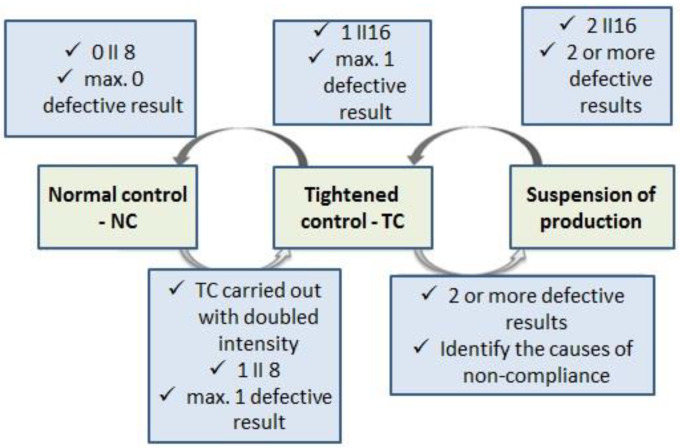
Conditions for the transition between the levels of control for the strength of concrete curbs (EN 1340:2004 [16]).

**Figure 12 materials-16-00431-f012:**
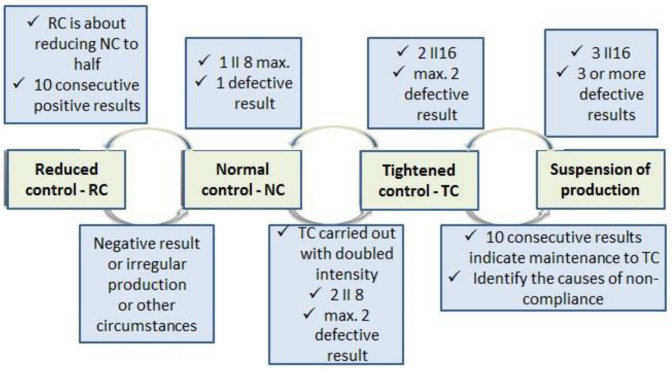
Transition conditions between the levels of control plans for assessing characteristics other than strength for concrete curbs (EN 1340:2004 [16]).

**Figure 13 materials-16-00431-f013:**
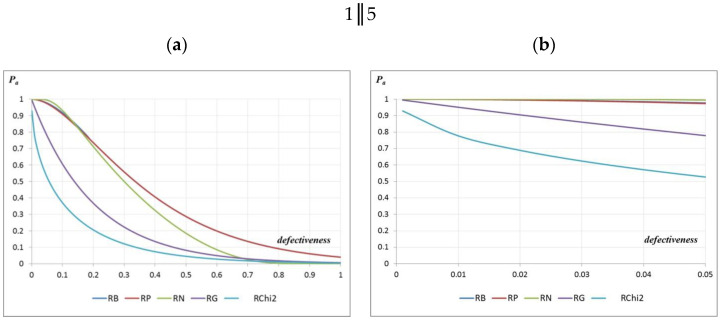
OC curves for testing plans: (**a**) 1‖5 and (**b**) 1‖5; 1‖16 (**c**) 1‖16 and (**d**) 1‖16.

**Figure 14 materials-16-00431-f014:**
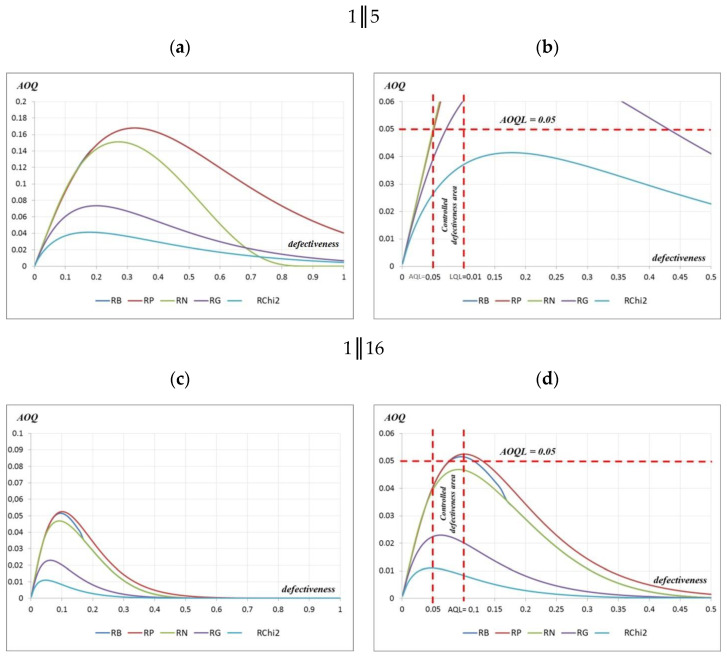
AOQ curves for testing plans: (**a**) 1‖5 and (**b**) 1‖5; 1‖16 (**c**) 1‖16 and (**d**) 1‖16.

**Figure 15 materials-16-00431-f015:**
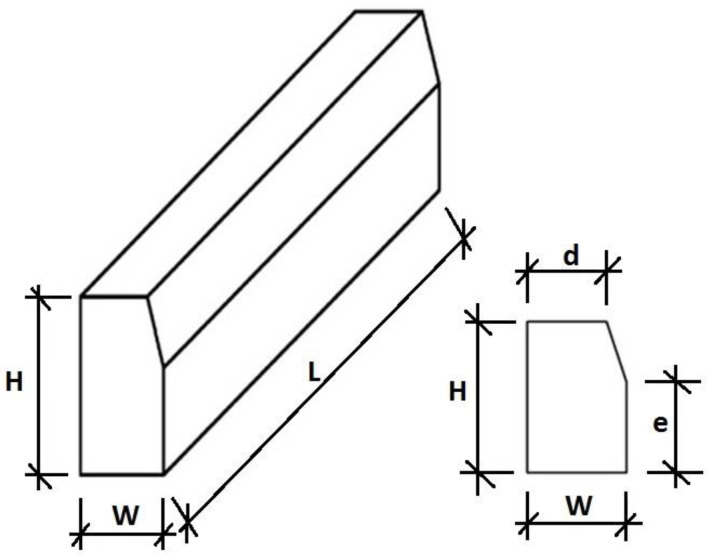
Description of the curbs’ geometry according to EN 1340:2004 [16].

**Figure 16 materials-16-00431-f016:**
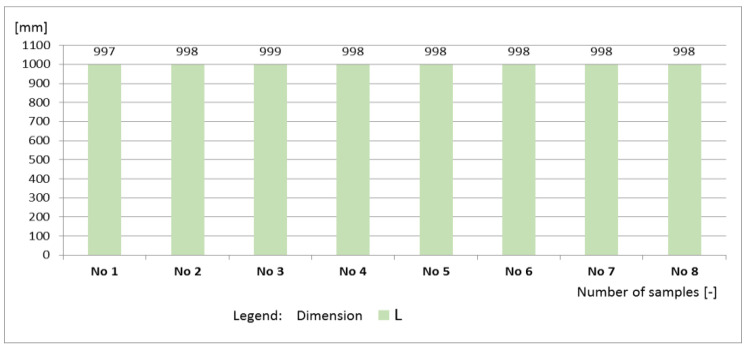
Values of L of the tested curbs.

**Figure 17 materials-16-00431-f017:**
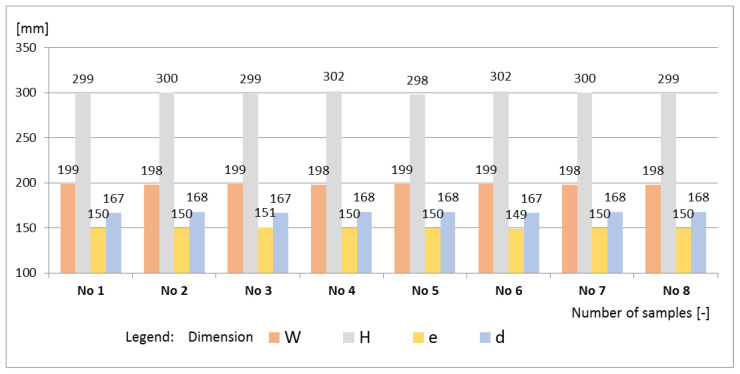
Values geometry of W, H, e, and d of the tested curbs.

**Figure 18 materials-16-00431-f018:**
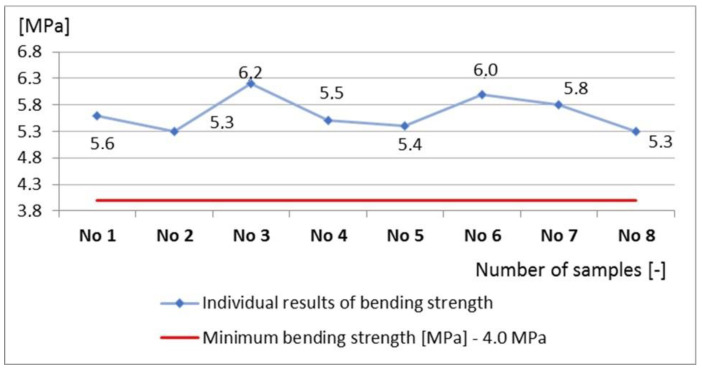
Bending strength values of the tested curbs.

**Figure 19 materials-16-00431-f019:**
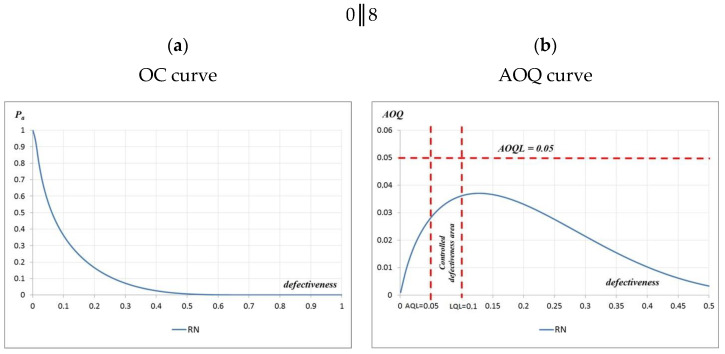
OC and AOQ curves for testing plan: (**a**) 0‖8 and (**b**) 0‖8.

**Table 1 materials-16-00431-t001:** The values of the acceptance probability estimated for various acceptance plans related to normal control and the transition from strict to normal control of the considered precast concrete products.

*A_c_*‖*n*	Probability of Acceptance—*P*(*w*) for Different Types of Distribution									
	RB	RP	RN	RG	RChi^2^	RB	RP	RN	RG	RChi^2^
	AQL = 0.05					AQL = 0.10				
0‖5	0.774	0.779	0.696	-	-	0.590	0.607	0.500	-	-
0‖8	0.773	0.670	0564	-	-	0.590	0.449	0.362	-	-
2‖5	0.998	0.997	0.999	0.974	0.819	0.991	0.986	0.998	0.910	0.670
1‖8	0.942	0.939	0.963	0.670	0.403	0.813	0.809	0.795	0.449	0.237
1‖16	0.811	0.808	0.789	0.448	0.221	0.515	0.525	0.466	0.202	0.083
2‖16	0.957	0.952	0.916	0.750	0.472	0.789	0.783	0.630	0.525	0.223

Legend: RB—Bernoulli distribution; RP—Poisson distribution; RN—normal distribution; RCh^2^—χ^2^ distribution; RG—Gamma distribution.

**Table 2 materials-16-00431-t002:** Number of rejected lots (1-P (w)) according to Table 1 for 1‖16 plan.

Defectiveness [–]	Probability of Acceptance for the 1‖16 Plan For Different Types of Distribution				
	RB	RP	RN	RG	RChi^2^
0.01	0.011	0.012	0.001	0.148	0.416
0.05	0.189	0.191	0.211	0.551	0.779
0.10	0.485	0.475	0.534	0.798	0.917

**Table 3 materials-16-00431-t003:** Producer’s and consumer’s risks for different plans.

*A_c_*‖*n*	Producer’s and Consumer’s Risks for Different Types of Distribution									
	RB	RP	RN	RG	RChi^2^	RB	RP	RN	RG	RChi^2^
	Producer’s Risk α (for a Given AQL = 0.05)					Consumer’s Risk β (for a Given LQL = 0.10)				
0‖5	0.226	0.221	0.304	-	-	0.590	0.607	0.500	-	-
0‖8	0.227	0.33	0.436	-	-	0.590	0.449	0.362	-	-
2‖5	0.002	0.003	0.001	0.026	0.181	0.991	0.986	0.998	0.910	0.670
1‖8	0.058	0.061	0.037	0.33	0.597	0.813	0.809	0.795	0.449	0.237
1‖16	0.189	0.192	0.211	0.552	0.779	0.515	0.525	0.466	0.202	0.083
2‖16	0.043	0.048	0.084	0.25	0.528	0.789	0.783	0.630	0.525	0.223

Legend: RB—Bernoulli distribution; RP—Poisson distribution; RN—normal distribution; RCh^2^—χ^2^ distribution; RG—Gamma distribution.

## Data Availability

Data sharing not applicable.

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
