# Peer review of "Statistical Quality Inspection Methodology in Production of Precast Concrete Elements"

_materials, 2023, doi:10.3390/ma16010431_

Round 1

Reviewer 1 Report

- References must be numbered in order of appearance in the text

  • Some paragraphs are extremely long, which can be challenging to read. Some rewriting work is suggested to shorten the paragraphs size.

Some minor writing errors are found in the article, which must be corrected.

- There are several times throughout the text a double space. Please correct.

Abstract:

"curves and AOQ (Average Outgoing Quality)curves." - it seems to be missing a space after the")" 

1. Introduction

"According to the research described in [ 30,25], the", there is an extra space after the "["

"factors and controlled parameters on the properties of output products.. However, in" - there is an "." more after "products"

"However, decisive in quality assurance is the correct implementation of the appropriate technology [26] Many companies" - it is missing one "." after "[26]" 

"The necessity to move from selective to full factory control has been found in tests, e.g. [44]" - it is missing one "." at the end of the sentence.

1.2. Development prospects for prefabrication in Poland 

- The first part of the paragraph incorrectly uses the Bold font style.

"Ease of assembly. is another factor that" - Replace the "." for a "," 

The phrase "It should be emphasized that" is repeatedly used, rewriting the paragraphs is suggested.

research are similar with the results of the research[48]. - missing a space before "[48]"

Author Response

The author wish to thank the Reviewer for the positive feedback and for all comments that helped to enrich and improve the paper. Thank you for all the suggestions.

The reviewer’s remarks and requests have been considered carefully by the author.

 All the requested revisions have been addressed. The author’s responses are presented below each of the Reviewer’s remarks.

  1. References must be numbered in order of appearance in the text

Ans. References have been corrected and numbered in the order in which they appear in the text.

  1. Some paragraphs are extremely long, which can be challenging to read. Some rewriting work is suggested to shorten the paragraphs size.

Ans.: The author is aware of lengthy paragraphs. However, the second Reviewer had no comments as to the length of individual paragraphs, so it was decided to leave individual paragraphs in their current form.

  1. Some minor writing errors are found in the article, which must be corrected.

 There are several times throughout the text a double space. Please correct.

Ans. The Reviewer’s remark has been considered. A double spaces in the text have been removed

 In Abstract:

 "curves and AOQ (Average Outgoing Quality)curves." - it seems to be missing a space after the")" 

Ans. The Reviewer’s remark has been considered. Added space after word “after”

  1. Introduction

 "According to the research described in [ 30,25], the", there is an extra space after the "["

 Ans. The Reviewer’s remark has been considered. The extra space has been removed

"factors and controlled parameters on the properties of output products.. However, in" - there is an "." more after "products"

 Ans. The Reviewer’s remark has been considered. A correction has been made.

"However, decisive in quality assurance is the correct implementation of the appropriate technology [26] Many companies" - it is missing one "." after "[26]" 

 Ans. The Reviewer’s remark has been considered.

 "The necessity to move from selective to full factory control has been found in tests, e.g. [44]" - it is missing one "." at the end of the sentence.

 Ans. The Reviewer’s remark has been considered.

1.2. Development prospects for prefabrication in Poland 

 - The first part of the paragraph incorrectly uses the Bold font style.

Ans. The Reviewer’s remark has been considered. Bold font removed.

 "Ease of assembly. is another factor that" - Replace the "." for a "," 

Ans. The Reviewer’s remark has been considered.

 The phrase "It should be emphasized that" is repeatedly used, rewriting the paragraphs is suggested.

Ans. The Reviewer’s remark has been considered. The phrase "It should be emphasized that" has been removed and modified.

research are similar with the results of the research[48]. - missing a space before "[48]"

Ans. The Reviewer’s remark has been considered. The missing space was added before [48].

 The author would like to thank the Reviewer

for all the helpful remarks and suggestions

Reviewer 2 Report

1. The abstract is to be made specific and clear.

2.The table format needs to be modified.

3.The font size in Figure 9 is inconsistent.

4.In section 3, it is suggested to give suggestions on the selection of sample quantity in different situations.

5. It is recommended to add actual cases to support the inspection results.

6.The conclusion needs to be refined to highlight the innovation and reference value of this article.

Author Response

Response to

the comments of Reviewer 2

The author wish to thank the Reviewer for the positive feedback and for all comments that helped to enrich and improve the paper. The author would like to especially thank the reviewer for all the suggested changes and raised queries directly marked on the paper.

The author wish to thank the Reviewer for the positive feedback and for all comments that helped to enrich and improve the paper. Thank you for all the suggestions.

The reviewer’s remarks and requests have been considered carefully by the author. All the requested revisions have been addressed. The author’s responses are presented below each of the Reviewer’s remarks.

  1. The abstract should be specific and clear.

The abstract has been revised. The author have attempted to underscore the scientific value added of the paper by pointing out to the application of statistical quality control method as a method used in solving in precast production problems, which are also scientific problems. In view of length limitations, the author have added the following sentence in the closing part of the abstract:

.In engineering practice, the recognition of the material's compliance with the specification is decided on the adopted plan for statistical quality control. It is a standard approach based on binary criteria (met/unmet). This is of particular significance in the case of the precast quality, where the precast quality is especially tightly linked with the structure's safety and reliability. The proposed procedure for the quality control of prefabricated elements allows for the evaluation of real quality of the precast produced. This was confirmed by the use of the proposed calculation algorithm for production data. The analyzes presented in the article can be used as a basis for further considerations regarding risk assessment in the production of prefabricated elements. They can be the basis for the evaluation of process quality control procedures and ready-made prefabricated elements, taking into account the economic boundary conditions.

  1. The table format needs modification.

Ans.: The author have made necessary corrections. The Table 7 has been revised.

  1. The font size in Figure 9 is inconsistent.

Ans.: The author have made necessary corrections. The Figure 9 has been revised.

  1. In section 3, it is suggested to give suggestions on the selection of sample quantity in different situations.

In section 3, suggestions on the selection of sample quantity in different situations were added.

In engineering practice, the recognition of the material's compliance with the specification is decided on the adopted plan for statistical quality control. It is a standard approach based on binary criteria (met/unmet). This is of particular significance in the case of doubts concerning the quality of precast elements, where the material quality is especially tightly linked with the structure's safety and reliability. When assessing quality, it is of great importance to adopt the appropriate sample quantity.

and

The numbers of samples recommended in the codes are the minimum values; therefore, the proposed statistical methods enable the adoption of the number of samples ensuring the safety and reliability of the constructed structures. The number of samples can be assumed based on the shape of the OC or AOQ curves and the type and reliability class of the building structures according to EN 1990 [66] i.e. RC1, RC2, RC3. According to the literature [50], the minimum statistically justified number of samples is 6.

  1. It is recommended to add actual cases to support the inspection results.

Ans. According to the reviewer's comment, to support the control results in section 2.3 added real case .

2.3. Case study of concrete kreb units

The data for the analysis were provided by Styrobud of Podkarpacie Province in Poland. Quality analysis was carried out for the continuous production of concrete kerb units. In accordance with the recommendations of the code EN 1340:2004 [16], the test covered 8 kerb units. The kerb units had production dimensions of 998x300x198 mm. Geometry and bending strength were evaluated. The description of the geometry of the element was adopted in accordance with the standard (Figure 15).

Figure 15. Description of the kerbs geometry

The kerbs belonged to quality class 2 (marking T), for which the characteristic bending strength is 5.0MPa and the minimum bending strength is 4.0MPa. An alternative method was used to evaluate the geometric dimensions and the bending strength. Each of the requirements specified in the standard regarding geometry and bending strength were met by all kerbs, thanks to which both the sample and the production lot from which it was taken were considered compliant with the requirements of the standard (Figures 16 and 17).

Figure 16. Values of L of the tested kerbs

Figure 17. Values geometry of W, H , e and d of the tested kerbs

No kerbs have a bending strength less than the characteristic value of the declared class T (Figure 18).

Figure 18. Bending strength values of the tested kerbs

An analysis of geometry and bending strength was also carried out for 32 results (four measurement cycles). The defectiveness was calculated for the geometry - w = 0.05 and for the bending strength of the tested kerbs - w = 0.03. Descriptive statistics for bending strength were determined: mean value – 5.3 MPa and standard deviation – 0.3MPa and then the characteristic value of the bending strength was determined, which was 5.0 MPa. Based on the determined parameters, the characteristic value of the bending strength was calculated, which was slightly higher than the minimum characteristic value specified in the standard 5.3 MPa > 5.0 MPa. Also, the obtained values ​​of defectiveness are lower than the limit values ​​of the quantiles given in the code EN 1340:2004 [16]  both for geometry w = 0.05 < 0.10 and for bending w = 0.03 < 0.05. The OC and AOQ curves were also constructed for the considered test plan 0â•‘8 and the acceptance probabilities of the tested kerbs were determined for the geometry Pa (0.05)=0.564 and for the bending strength Pa (0.05)=0.705 (Figure 19).

Figure 19. OC and AOQ curves for testing plan for 0â•‘8: OC curve - Figure 19a – and AOQ curve - Figure 19b.

The evaluated kerbs in terms of geometry and bending strength can therefore be considered as products that meet the requirements of the code EN 1340:2004 [16]. 

 6. The conclusion needs to be refined to highlight the innovation and reference value of this article

The innovation and reference value added of the paper has been underscored in the following paragraph added in the Conclusion section:

The traditional approach to the quality precast is through experiments and destructive testing, which, however, proves to be both time and resource consuming. The pro-posed statistical approach based method for quality assessment can overcome these limitations. The suggested method may be employed in precast production and applied to predict the quality of precast elements by reducing i.a. the numbers and scope of testing. The application of the proposed procedure combined with the use of methods: fuzzy,-statistical, fuzzy or an artificial neural networks and modern measurements technology can ensure the reliable assessment of precast quality.

 The author would like to thank the Reviewer

for all the helpful remarks and suggestions

Reviewer 3 Report

Dear Editor: 

This study does not contain any new ideas or contributions for further understanding of finding controlling parameters on the
precast concrete. Therefore, the main motivations behind this study should be comprehensively evaluated in the abstract, particularly in the introduction of this study with discussions and evaluations on the contributions, limitations, and weaknesses of similar previous studies. Otherwise, I should confess that I could not find any novelty in your study compared to the previous studies. The study lacks scientific novelty, the prediction and the data used are poor, and many grammar errors have negatively affected the quality of the paper.

Author Response

Response to the comments of Reviewer 3

The author wish to thank the Reviewer for all comments that helped to enrich and improve the paper. Thank you for all the suggestions.

The reviewer’s remarks and requests have been considered carefully by the author.

 This study does not contain any new ideas or contributions for further understanding of finding controlling parameters on the precast concrete. Therefore, the main motivations behind this study should be comprehensively evaluated in the abstract, particularly in the introduction of this study with discussions and evaluations on the contributions, limitations, and weaknesses of similar previous studies. Otherwise, I should confess that I could not find any novelty in your study compared to the previous studies. The study lacks scientific novelty, the prediction and the data used are poor, and many grammar errors have negatively affected the quality of the paper.

Ans.: The Reviewer’s remarks has been considered.

The abstract has been revised. The author have attempted to underscore the scientific value of the paper by pointing out to the application of statistical quality control methods with OC and AOQ curves as a  methods hitherto not used in solving in precast production problems, which are also scientific problems. The author has corrected:

  1. Introduction

Today, prefabricated concrete elements are used in many areas of construction, including in industrial, public, and residential construction; this was confirmed via questionnaire research. In this  article, the prospects for the development of precast concrete are presented, and the factors determining the use of this technology are defined. Based on a review of the literature, it was shown that currently, higher-quality prefabricated elements are primarily created through the implementation of innovative materials and production technologies. For this reason, the lack of research regarding quality control in prefabricated elements based on statistical quality control is particularly noticeable. The quality control process is one of the most important distinguishing features in prefabrication due to the increasingly stringent expectations of customers; it helps to ensure that the desired durability of implemented constructions is achieved. Issues related to the assessment of the effectiveness of standard procedures presented in this paper were analyzed using statistical methods in the form of OC (Operating Characteristic) and AOQ (Average Outgoing Quality) curves. Thus, a new approach was proposed, because these methods have not been previously used in precast concrete. The shape of the curves obtained confirmed the significant dependence of the value of the acceptance probability on the defectiveness of production. In AQL control systems based on OC and AOQ curves, it is necessary to calculate the current average defectiveness, which should be treated as a basis for the decision to switch from one type of control (normal, tightened, or reduced) to another. In this respect, the standard requirements of quality control have been simplified, and it has not been considered necessary to determine the average defectiveness value in production processes. The examples included in this study, including the analysis of curb production data, clearly show the negative effects of ignoring the actual process defectiveness. As a result of the calculations, it was found that the average actual defectiveness of the curbs produced could not be equated with batch defectiveness. The analyses carried out in this study prove that equating batch defectiveness with process defectiveness is not an appropriate approach, which was confirmed through the producer's/customer's risk analysis. The approach proposed in this study, the analysis of OC and AOQ curves, is an innovative solution in prefabrication and can be an effective tool for the management of the quality of prefabricated products, taking into account economic boundary conditions.

I also made changes to:

2.3. Case study of concrete curb units

  1. Results and discussion
  2. Conclusions

The author submitted the article for language correction to the English service center. Correction certificate is attached.

 The author would like to thank the Reviewer for all the helpful remarks and suggestions.

Reviewer 4 Report

·        The informal language is not suitable and should be improved extensively. The article needs major grammatical and syntax improvements. Use of English service center is recommended. Several sentences are not clear and understandable.

·        Majority of the qualitative statements should be modified for quantified result comparisons.    

·        The introduction needs to be revised for higher quality language. The author mentioned some works without stating about the contributions, pros and cons and the how the current work would address.

·        The purpose of the article should be clarified in details, why and where this study could be beneficent, more in depth conclusion should be provided.

·        The authors mentioned “Since concrete is the most widely used building material in the world.”  The following references should be added for comprehensiveness of this statement 1) Nano silica and metakaolin effects on the behavior of concrete containing rubber crumbs. CivilEng. 2) Investigation of steel fiber effects on concrete abrasion resistance, Advances in concrete construction.

·        Equation used previously should be clearly referenced.

·        More in depth conclusions should be drawn based on various studies, the summary should indicate in depth results and conclusions.

·        More descriptive legends and high quality figures are needed,

·        Any figures taken form other works should be reestablished and referenced.

·        Figures needs to be professionally done and caption should be more descriptive.

Author Response

The author would like to thank the Reviewer for all the helpful remarks and suggestions.

Response to the comments of Reviewer 4

The author wish to thank the Reviewer for the positive feedback and for all comments that helped to enrich and improve the paper. The author would like to especially thank the reviewer for all the suggested changes and raised queries directly marked on the paper.

The author wish to thank the Reviewer for the positive feedback and for all comments that helped to enrich and improve the paper. Thank you for all the suggestions.

The reviewer’s remarks and requests have been considered carefully by the author. All the requested revisions have been addressed. The author’s responses are presented below each of the Reviewer’s remarks.

  1. The informal language is not suitable and should be improved extensively. The article needs major grammatical and syntax improvements. Use of English service center is recommended. Several sentences are not clear and understandable.

The author submitted the article for language correction to the English service center. Correction certificate is attached.

  1. Majority of the qualitative statements should be modified for quantified result comparisons.

In Case study of concrete curb units and Results and discussion was supplemented with quantitative comparisons. Added among others Table 3 of Producer's and consumer's risks for different plans.

  1. The introduction needs to be revised for higher quality language. The author mentioned some works without stating about the contributions, pros and cons and the how the current work would address. The purpose of the article should be clarified in details, why and where this study could be beneficent, more in depth conclusion should be provided. The authors mentioned “Since concrete is the most widely used building material in the world.” The following references should be added for comprehensiveness of this statement

1) Nano silica and metakaolin effects on the behavior of concrete containing rubber crumbs. CivilEng.

2) Investigation of steel fiber effects on concrete abrasion resistance, Advances in concrete construction.

The introduction has been revised, references have been supplemented and the entire article has been linguistically corrected.

The author has corrected:

  1. Introduction

Today, prefabricated concrete elements are used in many areas of construction, including in industrial, public, and residential construction; this was confirmed via questionnaire research. In this  article, the prospects for the development of precast concrete are presented, and the factors determining the use of this technology are defined. Based on a review of the literature, it was shown that currently, higher-quality prefabricated elements are primarily created through the implementation of innovative materials and production technologies. For this reason, the lack of research regarding quality control in prefabricated elements based on statistical quality control is particularly noticeable. The quality control process is one of the most important distinguishing features in prefabrication due to the increasingly stringent expectations of customers; it helps to ensure that the desired durability of implemented constructions is achieved. Issues related to the assessment of the effectiveness of standard procedures presented in this paper were analyzed using statistical methods in the form of OC (Operating Characteristic) and AOQ (Average Outgoing Quality) curves. Thus, a new approach was proposed, because these methods have not been previously used in precast concrete. The shape of the curves obtained confirmed the significant dependence of the value of the acceptance probability on the defectiveness of production. In AQL control systems based on OC and AOQ curves, it is necessary to calculate the current average defectiveness, which should be treated as a basis for the decision to switch from one type of control (normal, tightened, or reduced) to another. In this respect, the standard requirements of quality control have been simplified, and it has not been considered necessary to determine the average defectiveness value in production processes. The examples included in this study, including the analysis of curb production data, clearly show the negative effects of ignoring the actual process defectiveness. As a result of the calculations, it was found that the average actual defectiveness of the curbs produced could not be equated with batch defectiveness. The analyses carried out in this study prove that equating batch defectiveness with process defectiveness is not an appropriate approach, which was confirmed through the producer's/customer's risk analysis. The approach proposed in this study, the analysis of OC and AOQ curves, is an innovative solution in prefabrication and can be an effective tool for the management of the quality of prefabricated products, taking into account economic boundary conditions.

I also made changes to:

2.3. Case study of concrete curb units

  1. Results and discussion
  2. Conclusions
  3. Equation used previously should be clearly referenced.

The equation references have been corrected.

  1. More in depth conclusions should be drawn based on various studies, the summary should indicate in depth results and conclusions.

The conclusions has been revised.

  1. More descriptive legends and high quality figures are needed. Any figures taken form other works should be reestablished and referenced. Figures needs to be professionally done and caption should be more descriptive.

Unreadable figures have been corrected. Figure 15 Description of the curbs geometry has been referenced to EN 1340:2004 [18]. Other drawings are own elaboration.

 The author would like to thank the Reviewer for all the helpful remarks and suggestions.

Round 2

Reviewer 3 Report

Dear Editor:

The authors carefully studied the reviewer's comments and revised the manuscript. In my opinion, this manuscript's quality meets the journal's requirements. I suggest this manuscript be accepted and published in this journal.

Author Response

I am grateful to Reviewer for his/her insightful review and comment. I made the 

The author wish to thank the Reviewer for all comments that helped to enrich and improve the paper.

As the Reviewer suggested, the author carefully checked English in terms of grammar, spelling, and syntax. A Native speaker has reviewed the article.

 The author would like to thank the Reviewer

for all the helpful remarks and suggestions

Reviewer 4 Report

·        The informal language is not suitable and should be improved. The article needs some grammatical and syntax improvements. 

· ·        The introduction needs to be revised for higher quality language. The author mentioned some works without stating about the contributions, pros and cons and the how the current work would address.

·        The authors mentioned “Since concrete is the most widely used building material in the world.”  The following references should be added for comprehensiveness of this statement 1) Compressive behavior of concrete under environmental effects. IntechOpen. 2) Temperature and humidity effects on behavior of grouts. Advances in concrete construction 3) Experimental investigation of sound transmission loss in concrete containing recycled rubber crumbs. 4) Nano silica and metakaolin effects on the behavior of concrete containing rubber crumbs. CivilEng. 5) Investigation of steel fiber effects on concrete abrasion resistance, Advances in concrete construction.

       Majority of the qualitative statements should be modified for quantified result comparisons.    

Author Response

I am grateful to Reviewer for his/her insightful review and comment. I made the article's subject and content fit into the journal's requirements and the research area that I carried out.

As the Reviewer suggested, the author carefully checked English in terms of grammar, spelling, and syntax. A Native speaker has reviewed the article.

The reviewer’s remarks and requests have been considered carefully by the author. All the requested revisions have been addressed. The author’s responses are presented below each of the Reviewer’s remarks.

 The author would like to thank the Reviewer

for all the helpful remarks and suggestions
